# Long-Term Resistance–Endurance Combined Training Reduces Pro-Inflammatory Cytokines in Young Adult Females with Obesity

**DOI:** 10.3390/sports11030054

**Published:** 2023-02-27

**Authors:** Adi Pranoto, Maulana Bagus Adi Cahyono, Reinaldi Yakobus, Nabilah Izzatunnisa, Roy Novri Ramadhan, Purwo Sri Rejeki, Muhammad Miftahussurur, Wiwin Is Effendi, Citrawati Dyah Kencono Wungu, Yoshio Yamaoka

**Affiliations:** 1Doctoral Program of Medical Science, Faculty of Medicine, Universitas Airlangga, Surabaya 60131, Indonesia; 2Medical Program, Faculty of Medicine, Universitas Airlangga, Surabaya 60131, Indonesia; 3Physiology Division, Department of Medical Physiology and Biochemistry, Faculty of Medicine, Universitas Airlangga, Surabaya 60131, Indonesia; 4Division of Gastroentero-Hepatology, Department of Internal Medicine, Faculty of Medicine, Universitas Airlangga, Dr. Soetomo Teaching Hospital, Institute of Tropical Disease, Surabaya 60131, Indonesia; 5Department of Pulmonology and Respiratory Medicine, Faculty of Medicine, Universitas Airlangga, Surabaya 60131, Indonesia; 6Biochemistry Division, Department of Medical Physiology and Biochemistry, Faculty of Medicine, Universitas Airlangga, Surabaya 60115, Indonesia; 7Department of Environmental and Preventive Medicine, Faculty of Medicine, Oita University, Yufu 879-5593, Japan

**Keywords:** cytokines, obesity, pro-inflammatory, sedentary lifestyle, combined training

## Abstract

A sedentary lifestyle and an unhealthy diet increase the risk of obesity. People with obesity experience adipocyte hypertrophy and hyperplasia, which increases the production of proinflammatory cytokines, thereby increasing the risk of morbidity and mortality. Lifestyle modification using non-pharmacological approaches such as physical exercise prevents increased morbidity through its anti-inflammatory effects. The purpose of this study was to examine the effects of different types of exercise on decreased proinflammatory cytokines in young adult females with obesity. A total of 36 female students from Malang City aged 21.86 ± 1.39 years with body mass index (BMI) of 30.93 ± 3.51 kg/m^2^ were recruited and followed three different types of exercise interventions: moderate-intensity endurance training (MIET), moderate-intensity resistance training (MIRT), and moderate-intensity combined training (MICT). The exercise was performed at a frequency of 3x/week for 4 weeks. Statistical analysis was performed using the Statistical Package for Social Science (SPSS) version 21.0, using the paired sample *t*-test. The results revealed that serum IL-6 and TNF-α levels were significantly decreased between pre-training and post-training in the three types of exercise (MIET, MIRT, and MICT) (*p* ≤ 0.001). The percentage change in IL-6 levels from pre-training in CTRL was (0.76 ± 13.58%), in MIET was (−82.79 ± 8.73%), in MIRT was (−58.30 ± 18.05%), in MICT was (−96.91 ± 2.39%), and (*p* ≤ 0.001). There was a percentage change in TNF-α levels from pre-training in CTRL (6.46 ± 12.13%), MIET (−53.11 ± 20.02%), MIRT (−42.59 ± 21.64%), and MICT (−73.41 ± 14.50%), and (*p* ≤ 0.001). All three types of exercise consistently reduced proinflammatory cytokines such as serum levels of IL-6 and TNF-α.

## 1. Introduction

Obesity has increased globally over the last 50 years and is now considered an epidemic or even a pandemic in the 21st century [1,2]. Currently, the prevalence of obesity worldwide has more than doubled compared to 1975, when only 3.2% of men and 6.4% of women had obesity. If this pattern of growth continues, the prevalence of obesity will inevitably increase to 18% in men and 21% in women by 2025 [3]. According to Indonesian Basic Health Research (RISKESDAS) 2018, the prevalence of obesity among people older than 18 years was 21.8%, which was higher than in 2013 (14.8%) and 2007 (10.5%) [4]. Notably, this condition is alarming and poses a serious threat to the health of populations worldwide [5,6].

Physiologically, obesity is associated with low-grade chronic inflammation (LGCI) [7]. This is because individuals with obesity experience adipocyte hypertrophy and hyperplasia [8], causing a mechanical stressor that stimulates macrophage M1 activation [9]. Proinflammatory cytokines such as tumor necrosis factor-alpha (TNF-α) and interleukin 6 (IL-6) are released by activated M1 macrophages [10]. The implication of this is an increased risk of insulin resistance [11], myocardial infarction [12], hypertension, stroke [13], gallstones, non-alcoholic fatty liver disease (NAFLD) [14], and several types of cancer [15,16,17,18]. Moreover, the onset of obesity often occurs in the young population, and this population will likely suffer from more extended morbidity [19]. An unhealthy diet and lack of physical activity lead to obesity [6,20,21]. Through their anti-inflammatory effects, lifestyle changes using an exercise-based non-pharmacological approach are considered effective in preventing increased morbidity [22,23].

Regular moderate-intensity exercise has an anti-inflammatory effect by lowering blood levels of basal cytokines such as IL-6 and TNF-α [16,24]. In women with obesity, hypertrophy and hyperplasia of adipocytes trigger stress and stimulate macrophage activation [25,26]. Activated macrophages induce the production of IL-6 and TNF-α via the nuclear factor-kappa beta (NF-κβ) pathway [27,28]. Regular exercise induces increased IL-6 secretion into the blood [22], resulting in the activation of the extracellular signal-regulated kinase 1/2 (ERK_1/2_) pathway, thereby increasing lipolysis [29,30,31]. Continuous lipolysis induced by regular exercise can reduce the amount of visceral fat in the body [32], leading to lower basal levels of serum IL-6 and TNF-α [33]. Exercise can also cause muscles to adapt by increasing the number of mitochondria and becoming more efficient at oxidizing fatty acids, thereby reducing the need for glycogen, blood glucose, and lactate production during exercise [34,35,36]. A decrease in glycogen correlates with a decrease in IL-6 synthesis through the AMP-activated protein kinase (AMPK) and p38 mitogen-activated protein kinase (p38MAPK) pathways [37]. Exercise also correlates with decreased M1 infiltration of adipocytes such that proinflammatory cytokine levels at basal levels decrease [38,39]. Previous studies have revealed that in comparison to resistance training, endurance exercise was more effective at lowering the blood levels of IL-6 and TNF-α [40]. Other studies have revealed that serum levels of IL-6 and TNF-α were higher in endurance exercise than in resistance exercise [41]. Based on the results of these studies, there remains an uncertainty regarding the different types of exercise that may be used to reduce IL-6 and TNF-α levels. Appropriate exercise interventions can be a solution for reducing the level of inflammation and mortality and morbidity rates among patients with obesity. According to the latest guidelines by the World Health Organization and the American College of Sports Medicine, regular resistance training can provide global health benefits by preventing cardiovascular disease mortality, incident hypertension, type 2 diabetes, and cancers, and improving mental health (by reducing symptoms of depressive and anxiety), and can also improve adiposity [42,43,44]. Therefore, the purpose of this study was to examine the effects of different types of exercise on decreased proinflammatory cytokines in young adult females with obesity.

## 2. Materials and Methods

### 2.1. Research Design

This was a true experimental study with a pre-test–post-test control group design. A total of 42 participants underwent anthropometric measurements and body composition. Six participants did not meet the inclusion criteria. A total of 36 women with obesity (based on Asia-Pacific BMI classification) who had a body mass index (BMI) of 30.93 ± 3.51 kg/m^2^, an age of 21.86 ± 1.39 years, blood pressure of (systolic blood pressure 113.50 ± 9.52 mmHg, diastolic blood pressure 79.92 ± 7.37 mmHg), a resting heart rate of (RHR) 79.89 ± 7.97 bpm, oxygen saturation (SpO_2_) 97.58 ± 1.96%, a fasting blood sugar of 91.72 ± 5.84 mg/dL, and hemoglobin of 15.24 ± 1.94 g/dL were selected to be participants in the study, and blood samples were collected. All participants were provided information about the study orally and in writing, and they consciously filled out and signed an informed consent form. All selected respondents had no history of smoking, alcohol consumption, or chronic diseases such as kidney failure, lung disease, diabetes mellitus, hypertension, or cardiovascular disease. It was confirmed that all respondents were not undergoing a weight loss program using either medication or diet, and were not active in sports activities. The participants were randomly assigned to four groups: CTRL (*n* = 9, control group), MIET (*n* = 9, moderate-intensity endurance training group), MIRT (*n* = 9, moderate-intensity resistance training group), and MICT (*n* = 9, moderate-intensity combined training group). The exercise was performed at a frequency of 3x/week for 4 weeks. In this study, the dropout rate was 0%, and the attendance rate was 100%. After the 4 week intervention program, anthropometric and body composition measurements were taken, and blood samples were collected. A flowchart of the study is shown in Figure 1.

### 2.2. Measurement of Body Composition and Physiological Parameters

Height was measured using a portable stadiometer Seca 213. Body composition was measured using a Seca mBCA 554. Seca Mbca 554 is a medical body composition analyzer that uses BIA to calculate body composition. Blood pressure and resting heart rate were measured using an OMRON HBP-9030 digital tensiometer. A Beurer PO 30 Pulse Oximeter was used to measure oxygen saturation (SpO_2_), while body temperature (BT) was measured using an OMRON MC-343F Digital Thermometer administered orally.

### 2.3. Exercise Protocol and Blood Sampling

The exercise protocol was implemented and supervised by a personal trainer from Atlas Sports Club Malang (East Java 65146, Indonesia). The MIET intervention was performed by running on a treadmill at an intensity of 60–70% HRmax for 35 min. The MIRT intervention was performed with an intensity of 60–70% 1-RM, with 4–6 sets at 12–15 reps, with active rest of 30 s between sets, for a total exercise time of 35 min/session. The method used in MIRT is circuit training, divided into upper and lower body parts. Upper body resistance training included pull-downs, shoulder presses, chest presses, and tricep push-downs, while lower body resistance training included leg presses, leg extensions, leg curls, and barbell squat presses (upper and lower body resistance training was not performed simultaneously, but on different days). The MICT was performed by combining endurance training with resistance training on separate days (lower body resistance training (Monday), upper body resistance training (Wednesday), and endurance training (Friday)) with a training duration of 35 min/session. The warm-up and cool-down for the three types of exercise were each performed for 5 min at an intensity of 50% HRmax and carried out by walking quickly on the treadmill. The exercise was performed at a frequency of 3x/week for 4 weeks. The intervention was conducted from 07.00–09.00 am in Fitness Center Atlas Sports Club Malang (East Java 65146, Indonesia). Heart rate during exercise was monitored using a Polar H10 heart rate sensor. The study environment had a humidity level of 50–70% and a room temperature of 26 ± 1 °C [45]. Blood sampling was carried out twice, both pre-training (0 weeks) and 24 h post-training (4 weeks) in the cubital vein (4 mL). Blood sampling both pre-and post-training was carried out after the subjects had fasted overnight for 12 h. The collected blood samples were centrifuged for 15 min at 3000 rpm. The separated serum samples were immediately processed to analyze IL-6 and TNF-α levels.

### 2.4. Measurement of Cytokines Pro-Inflammatory

Pro-inflammatory cytokines such as IL-6 levels were measured using commercial ELISA Kits (Cat.No.:E-EL-H6156; Human IL-6 ELISA Kit; Elabscience Biotechnology Inc., Houston, TX 77079, USA) with a sensitivity level of 0.94 pg/mL and a detection range of 1.56–100 pg/mL, while TNF-α levels were measured using commercial ELISA kits (Cat.No.: E-EL-H0109; Human TNF-α ELISA Kit; Elabscience Biotechnology Inc., Houston, TX 77079, USA) with a sensitivity level of 4.69 pg/mL and a detection range of 7.81–500 pg/mL. Several studies have validated the accuracy of commercial ELISA kits used to analyze IL-6 and TNF-α levels [22,46,47].

### 2.5. Statistical Analysis

Normality was assessed using the Shapiro–Wilk test, while Levene’s test was used for homogeneity. The parametric paired sample *t*-test was used to determine the differences in data between pre- and post-training in each group. The data were normally distributed, and a one-way ANOVA test was performed to determine the difference in data in all groups, followed by Tukey’s HSD post hoc test. If the data were not normally distributed, the non-parametric Kruskal–Wallis Test was performed, followed by the Mann–Whitney U test. The relationship between the parameters was evaluated using Pearson’s correlation coefficient test. All statistical analyses used a significance level of 5%, and all data are presented as mean ± SD.

## 3. Results

In this study, the dropout rate was 0%, and the attendance rate was 100%. The analysis results from the basic characteristics of all subjects in the four groups shown in Table 1. The analysis results of the average levels of pro-inflammatory cytokines based on time in the four groups (CTRL vs. MIET vs. MIRT vs. MICT) shown in Table 2.

The analysis of the average pre-training serum IL-6 levels in the four groups (CTRL, MIET, MIRT, MICT) showed no significant difference (*p* ≥ 0.05), whereas the non-parametric analysis of the average post-training serum IL-6 levels and delta (Δ) showed a significant decrease in all four groups (CTRL, MIET, MIRT, MICT) (*p* ≤ 0.001). A significant difference (*p* ≤ 0.001) in the average decrease in post-training serum IL-6 levels between MICT vs. CTRL, MICT vs. MIET, MICT vs. MIRT, MIET vs. CTRL, MIET vs. MIRT, and MIRT vs. CTRL is shown in the Mann–Whitney U Test. The non-parametric analysis also showed a significant difference in the average delta (Δ) decrease in serum IL-6 levels between MICT vs. CTRL, MICT vs. MIRT, MIET vs. CTRL, and MIRT vs. CTRL, with significant values (*p* ≤ 0.001), while the average delta (Δ) decreased serum IL-6 levels between MICT vs. MIET, and MIET vs. MIRT showed no significant difference (*p* ≥ 0.05).

The analysis of the average pre-training serum TNF-α levels in the four groups (CTRL, MIET, MIRT, MICT) showed no significant differences (*p* ≥ 0.05), while the results of the non-parametric analysis on the average post-training serum TNF-α levels and delta (Δ) showed a significant decrease in all four groups (CTRL, MIET, MIRT, MICT) (*p* ≤ 0.001). The non-parametric analysis also showed a significant difference in the average decrease in post-training serum TNF-α levels between MICT vs. CTRL, MICT vs. MIET, MICT vs. MIRT, MIET vs. CTRL, and MIRT vs. CTRL, with significant values (*p* ≤ 0.001), while MIET and MIRT showed no significant difference in the average decrease in serum TNF-α levels post-training (*p* ≥ 0.05). The non-parametric analysis also showed a significant difference in the average delta (Δ) decrease in serum TNF-α levels between MICT vs. CTRL, MIET vs. CTRL, and MIRT vs. CTRL with significant values (*p* ≤ 0.001), while the average delta (Δ) decreased serum TNF-α levels between MICT vs. MIET, and MIET vs. MIRT showed no significant difference (*p* ≥ 0.05).

The analysis in Table 3 shows a positive relationship between Δ serum IL-6 levels and Δ body mass index, Δ body fat percentage, Δ fat mass, Δ fat-free mass, Δ waist circumference, Δ waist-to-hip ratio, and Δ leptin, and showed a moderate correlation among variables. Δ serum IL-6 levels were also found to have a negative relationship with Δ skeletal muscle mass and Δ adiponectin, and showed a moderate correlation between the three. Δ serum TNF-α levels were found to have a positive relationship with Δ body mass index, Δ body fat percentage, Δ waist circumference, Δ waist-to-hip ratio, and Δ leptin, and showed a moderate correlation among variables. However, other body fat markers such as Δ fat mass and Δ fat-free mass showed weak positive correlations with Δ serum TNF-α levels. Δ serum TNF-α levels were also found to be negatively correlated with Δ skeletal muscle mass and Δ adiponectin, and showed a moderate correlation between the three.

## 4. Discussion

Our study showed that there was no significant difference in serum IL-6 and TNF-α levels between the CTRL pre-training and post-training groups, whereas MIET, MIRT, and MICT showed significant decreases in serum IL-6 and TNF-α levels pre-and post-training (Figure 1 and Figure 2). However, the analysis revealed that the greatest decrease occurred in MICT, followed by MIET and MIRT (Figure 2 and Figure 3). Combination training is considered the best exercise because it has the most anti-inflammatory effect compared to aerobic exercise and resistance training [48,49]. Combination exercise reduces serum IL-6 and TNF-α levels more effectively than aerobic or resistance training [50]. Our study results reported that combined training is better at reducing pro-inflammatory cytokines than aerobic and resistance training independently. In this study, aerobic exercise was considered better than resistance training at reducing serum IL-6 and TNF-α levels. This is in accordance with a previous study by Sabag et al. [40], which stated that aerobic exercise for 45 min with a capacity of 50–70% HRmax for 12 weeks in female participants with obesity aged 50–60 years resulted in a greater decrease in IL-6 levels than resistance training. Ho et al. [48] reported that 12 weeks of aerobic exercise in participants with obesity aged 40–66 years at an intensity of 60% HRR for 30 min was more effective in lowering serum TNF-α levels than 12 weeks of resistance training in four sets of 8–12 repetitions.

A meta-analysis of the obese population related to cardiometabolic health parameters revealed that combined training is more effective in controlling body composition, cardio-respiratory fitness, blood lipids, blood pressure, and blood glucose compared with endurance training and resistance training [51]. Combined training integrates aerobic and muscle-strengthening activities into a single session. This multi-component exercise modality has a greater effect on increasing the metabolic rate, FFM, high-density lipoprotein (HDL,) and insulin while also lowering SBP, DBP, mean arterial pressure (MAP), and low-density lipoprotein (LDL). Hence, currently, international guidelines recommend systematic multicomponent exercise regimens for obese individuals.

Adipocyte hypertrophy and hyperplasia in women with obesity causes stress and activation of macrophages. Activated macrophages release TNF-α and induce IL-6 production via the NF-kβ pathway [27,28]. MIET inhibits adipocyte hypertrophy and hyperplasia via lipolysis [24]. Lipolysis occurs via the ERK_1/2_ pathway, which is activated after a single session of aerobic exercise [52]. Regular exercise contributes to this mechanism, resulting in a reduction in fat mass [32] and inflammatory markers. MIRT suppresses IL-6 and TNF-α production through muscle adaptation to stressors. Muscle contraction during exercise reduces glycogen reserves in muscles. Reduced glycogen reserves promote myokine IL-6 production via the AMPK and p38 AMPK pathways. Activation of this pathway triggers an increase in IL-6 mRNA levels in muscle and blood. Elevated serum IL-6 signals the liver to initiate gluconeogenesis. MIRT regularly adapts muscles by increasing the number of mitochondria in muscles and making fatty acid oxidation more efficient, thus reducing the need for glycogen, blood glucose, and lactate production during exercise [37]. MIRT has also been shown to lower TNF-α gene expression [53]. The increase in beta-oxidation and lipase enzymes after MIRT enhances fatty acid uptake in the skeletal muscles, which directly lowers adipose tissue [54]. Consequently, the levels of inflammatory markers (IL-6 and TNF-α) decrease. Therefore, combined exercise has the best results in terms of lowering inflammatory markers (IL-6 and TNF-α), because it incorporates both the MIET mechanism that stimulates lipolysis and the MIRT mechanism that enables muscle adaptation to stressors.

Adipocytes produce inflammatory markers such as IL-6 and TNF-α [55]. This is evidenced by the correlation between body fat mass and inflammatory markers. A meta-analysis of patients with obesity who participated in an exercise intervention for at least 4 weeks discovered that aerobic exercise burns visceral fat more effectively than resistance training or combination therapy. Resistance training and post-exercise combinations did not significantly change visceral fat levels [50]. However, this might be influenced by the participant’s diet, which cannot be controlled in the study; therefore, it is possible that participants from certain groups consumed foods with high fat content. Based on previous studies, it is possible to hypothesize that the decrease in fat mass, as confirmed in this study, was a determinant of the reduction in serum levels of IL-6 and TNF-α in young women with obesity [49].

Regular exercise can reduce morbidity and mortality, especially when started at a young age [15]. Increased levels of inflammatory markers (IL-6 and TNF-α) in the body are associated with an increased risk of insulin resistance and cardiovascular disease [56]. Due to the increased maximum oxygen volume (VO_2_max), exercise increases the expression of glucose transporter type 4 (GLUT4) in cell membranes, improving insulin sensitivity by 25–50% and helping in maintaining normal blood glucose levels [57]. Therefore, exercise can be one of the pillars of lifestyle modification in individuals with obesity [21,45]. Regular exercise can reduce stenosis in coronary arteries by increasing nitric oxide (NO) and endothelial nitric oxide synthase (eNOS) production, and increasing anti-inflammatory cytokines in the vascular endothelium [58].

Strict inclusion criteria were used, and preliminary research was conducted to measure energy expenditure between types of exercise; therefore, the exercise was expected to provide good results and form a non-pharmacological regimen for obese people. This study has a few limitations. First, the study was limited to IL-6 and TNF-α as parameters; thus, we were unable to explain the molecular mechanisms involved in reducing IL-6 and TNF-α levels. Second, this study was only conducted on young adult females with obesity; therefore, the findings of this study cannot be generalized to all age groups and sexes. Another limitation is the length of the intervention. The data cannot be extrapolated to any time frame outside of 4 weeks. In addition, we could not fully control any physical activity carried out outside the 4-week training program. During the intervention period, we were unable to control the participants’ food intake. To prove the mechanism, this study needs further research to explore the molecular mechanisms underlying the reduction of pro-inflammatory cytokines such as IL-6 and TNF-α.

## 5. Conclusions

The findings of this study showed that all three types of exercise reduced the levels of proinflammatory cytokines IL-6, and TNF-α. However, moderate-intensity combination exercise was more optimal in lowering these proinflammatory cytokines than endurance exercise and resistance training, based on the difference within observation time. Further research might be conducted in a larger population including male patients with obesity and for a longer duration (more than 4 weeks) in order to produce more precise results.

## Figures and Tables

**Figure 1 sports-11-00054-f001:**
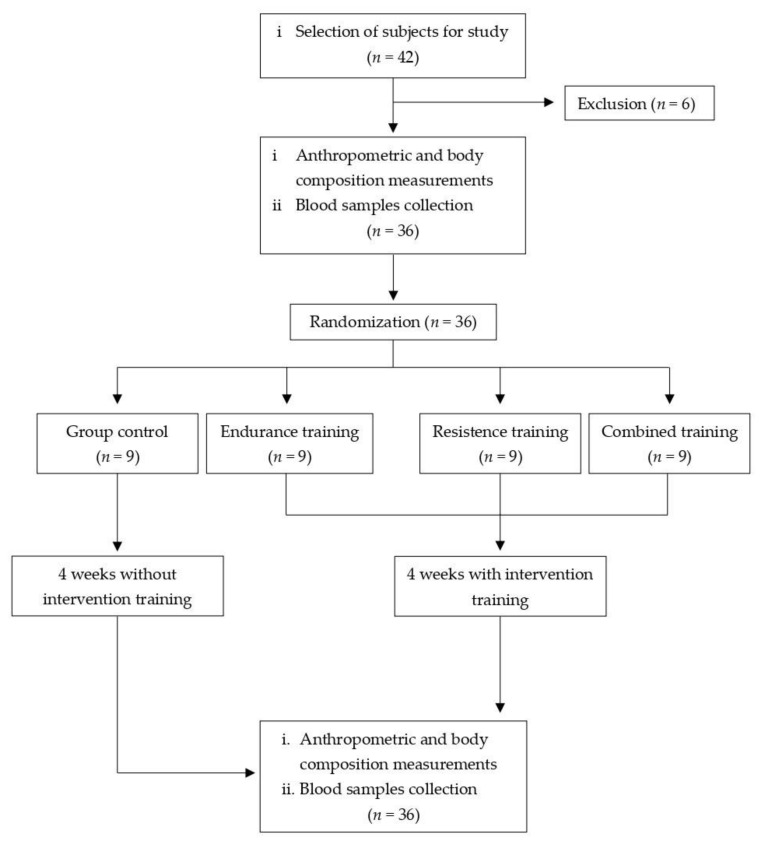
Flowchart of the study.

**Figure 2 sports-11-00054-f002:**
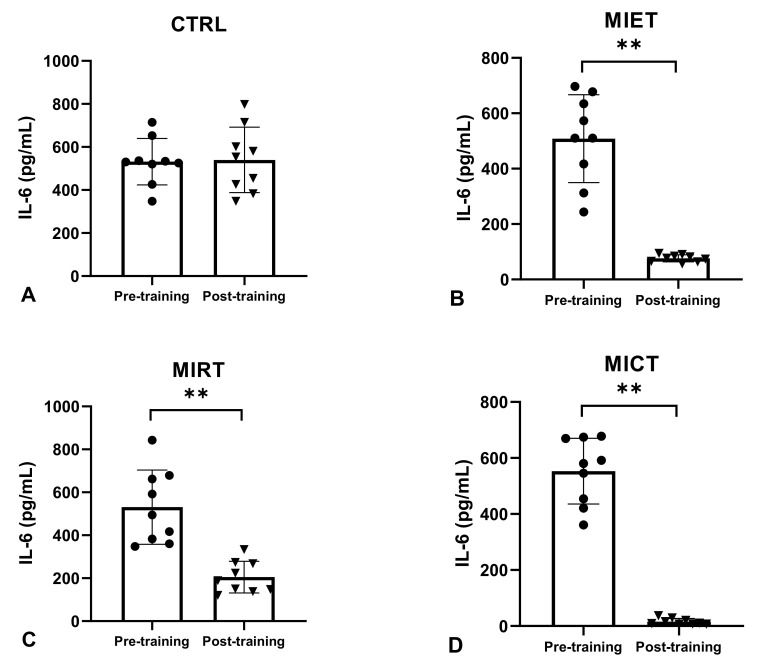
The average levels of IL-6 between pre-training and post-training in the four groups (CTRL, MIET, MIRT, MICT). Description: (**A**) CTRL: Control group; (**B**) MIET: Moderate-intensity endurance training group; (**C**) MIRT: Moderate-intensity resistance training group; (**D**) MICT: Moderate-intensity combined training group. (**) Significant with pre-training in all three intervention groups (MIET, MIRT, and MICT) (*p* ≤ 0.001). Values are expressed as mean ± SD. A paired sample *t*-test was used to determine the *p*-values.

**Figure 3 sports-11-00054-f003:**
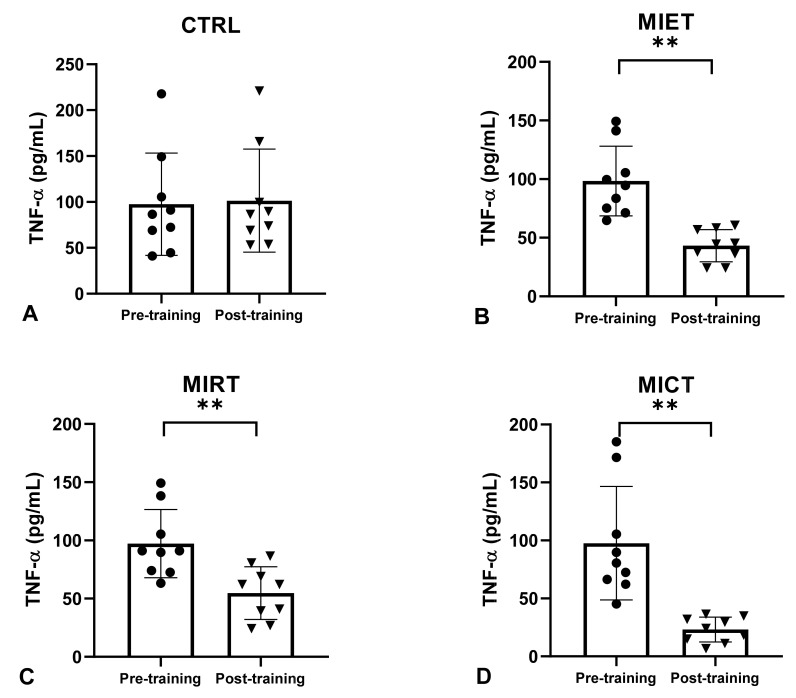
The average levels of TNF-α between pre-training and post-training in the four groups (CTRL, MIET, MIRT, MICT). Description: (**A**) CTRL: Control group; (**B**) MIET: Moderate-intensity endurance training group; (**C**) MIRT: Moderate-intensity resistance training group; (**D**) MICT: Moderate-intensity combined training group. (**) Significant with pre-training in all three intervention groups (MIET, MIRT, and MICT) (*p* ≤ 0.001). Values are expressed as mean ± SD. A paired sample *t*-test was used to determine the *p*-values.

**Table 1 sports-11-00054-t001:** The basic characteristics of all subjects in the four groups (CTRL, MIET, MIRT, MICT).

Parameter	(CTRL; n = 9)	(MIET; n = 9)	(MIRT; n = 9)	(MICT; n = 9)	*p*-Value
Age, yrs	22.22 ± 1.64	21.44 ± 1.67	21.78 ± 1.30	22.00 ± 1.00	0.695
BH, m	1.59 ± 0.06	1.55 ± 0.07	1.56 ± 0.06	1.57 ± 0.05	0.910
BW, kg	77.47 ± 10.06	74.13 ± 8.32	76.20 ± 9.98	76.51 ± 11.30	0.659
BMI, kg/m^2^	30.69 ± 4.01	30.64 ± 3.39	31.20 ± 3.58	31.19 ± 3.61	0.979
FM, kg	35.25 ± 6.42	32.49 ± 5.12	34.57 ± 6.71	32.99 ± 6.34	0.755
FM, %	39.48 ± 7.46	43.63 ± 2.43	45.11 ± 3.94	42.70 ± 3.36	0.095
FMI, kg/m^2^	13.03 ± 2.79	13.43 ± 2.12	14.24 ± 2.85	13.36 ± 2.23	0.770
FFM, kg	44.69 ± 3.96	41.65 ± 3.47	41.72 ± 4.79	43.95 ± 5.58	0.377
FFMI, kg/m^2^	17.77 ± 1.43	17.20 ± 1.42	17.07 ± 1.08	17.82 ± 1.74	0.587
SMM, kg	19.91 ± 2.56	18.44 ± 2.05	19.94 ± 4.69	20.33 ± 3.19	0.632
TBW, %	32.88 ± 3.21	30.60 ± 2.56	31.43 ± 4.81	32.52 ± 4.18	0.572
ECW, %	15.00 ± 1.50	13.79 ± 1.26	14.20 ± 1.99	14.40 ± 1.83	0.491
ECW/TBW, %	45.72 ± 1.39	45.21 ± 1.31	45.27 ± 1.29	44.27 ± 0.86	0.106
TEE, kcal/day	2654.22 ± 149.78	2590.33 ± 153.71	2627.22 ± 181.74	2630.33 ± 204.23	0.891
REE, kcal/day	1551.78 ± 99.27	1523.78 ± 90.37	1545.44 ± 106.78	1547.33 ± 120.23	0.943
WC, m	0.93 ± 0.09	0.92 ± 0.11	0.91 ± 0.04	0.90 ± 0.05	0.792
HC, m	1.11 ± 0.09	1.12 ± 0.16	1.09 ± 0.05	1.14 ± 0.19	0.928
WHR	0.84 ± 0.06	0.83 ± 0.08	0.83 ± 0.05	0.80 ± 0.09	0.694
SBP, mmHg	115.67 ± 8.09	109.33 ± 8.41	114.89 ± 10.79	114.11 ± 10.77	0.507
DBP, mmHg	78.22 ± 8.01	78.67 ± 7.48	82.11 ± 8.41	80.67 ± 5.96	0.671
RHR, bpm	77.44 ± 9.51	76.56 ± 6.56	83.56 ± 7.58	82.00 ± 6.87	0.174
SpO_2_, %	97.89 ± 0.93	96.67 ± 3.39	98.11 ± 1.05	97.67 ± 1.41	0.430
BT, °C	36.21 ± 0.36	36.24 ± 0.22	36.28 ± 0.23	36.13 ± 0.18	0.666
FBS, mg/dL	90.89 ± 6.53	91.22 ± 5.85	91.00 ± 6.34	93.78 ± 5.07	0.700
Hb, g/dL	14.21 ± 2.70	15.44 ± 1.28	15.57 ± 1.34	15.75 ± 2.01	0.326
Baseline IL-6 (pg/mL)	531.93 ± 108.02	508.22 ± 158.55	530.90 ± 172.92	552.82 ± 117.37	0.930
Baseline TNF-α (pg/mL)	97.51 ± 55.64	98.35 ± 29.82	97.20 ± 29.35	97.65 ± 48.92	0.990

Description: BH: Body height; BMI: Body mass index; BT: Body temperature; BW: Body weight; CTRL: Control group; DBP: Diastolic blood pressure; ECW: Extracellular water; ECW/TBW: Extracellular water/total body water; FBS: Fasting blood sugar; FFM: Free fat mass; FFMI, free fat mass index; FM: Fat mass; FMI: Fat mass index; Hb: Hemoglobin; MICT: Moderate-intensity combined training group; MIET: Moderate-intensity endurance training group; MIRT: Moderate-intensity resistance training group; REE: Resting energy expenditure; RHR: Resting heart rate; SBP: Systolic blood pressure; SMM: Skeletal muscle mass; SpO_2_: Oxygen saturation; TBW: Total body water; TEE: Total energy expenditure. Values are expressed as mean ± SD. One-way ANOVA was used to determine the *p*-values.

**Table 2 sports-11-00054-t002:** The average levels of pro-inflammatory cytokines based on time in the four groups (CTRL vs. MIET vs. MIRT vs. MICT).

Time	(CTRL; n = 9)	(MIET; n = 9)	(MIRT; n = 9)	(MICT; n = 9)	*p*-Value
Pre-training IL-6 (pg/mL)	531.93 ± 108.02	508.22 ± 158.55	530.90 ± 172.92	552.82 ± 117.37	0.930 †
Post-training IL-6 (pg/mL)	539.90 ± 151.91	76.20 ± 12.61 **&	205.00 ± 74.08 **	15.92 ± 10.82 **$&	0.000 #
Δ IL-6 (pg/mL)	7.97 ± 70.93	−432.02 ± 163.76 **	−325.90 ± 170.31 **	−536.90 ± 118.74 **&	0.000 #
IL-6 percent changes from pre-training (%)	0.76 ± 13.58	−82.79 ± 8.73 **$	−58.30 ± 18.05 **	−96.91 ± 2.39 **$&	0.000 #
Pre-training TNF-α (pg/mL)	97.51 ± 55.64	98.35 ± 29.82	97.20 ± 29.35	97.65 ± 48.92	0.990 †
Post-training TNF-α (pg/mL)	101.44 ± 56.12	43.17 ± 13.78 **	54.73 ± 22.71 *	23.17 ± 10.80 **$&	0.000 #
Δ TNF-α (pg/mL)	3.93 ± 7.71	−55.17 ± 29.95 **	−42.47 ± 29.29 **	−74.48 ± 45.15 **	0.000 #
TNF-α percent changes from pre-training (%)	6.46 ± 12.13	−53.11 ± 20.02 **	−42.59 ± 21.64 **	−73.41 ± 14.50 **&	0.000 †

Description: (*) Significant with CTRL (*p* ≤ 0.05). (**) Significant difference with CTRL (*p* ≤ 0.001). ($) Significant for MIET (*p* ≤ 0.001). (&) Significant with MIRT (*p* ≤ 0.001). (†) *p*-value determined by one-way parametric ANOVA and followed by Tukey’s HSD post hoc test. (#) *p*-value determined using the Kruskal–Wallis non-parametric test and continued with the Mann–Whitney U Test. Values are expressed as mean ± SD.

**Table 3 sports-11-00054-t003:** The association between proinflammatory cytokines with body fat and adipokine.

Parameter	Pro-inflammatory cytokines			
	Δ IL-6 (pg/mL)		Δ TNF-α (pg/mL)	
	*R*	*p*-Value	*r*	*p*-Value
Δ Body mass index, kg/m^2^	0.622 **	*p* ≤ 0.001	0.576 **	*p* ≤ 0.001
Δ Fat mass, kg	0.728 **	*p* ≤ 0.001	0.481 **	*p* ≤ 0.001
Δ Body fat percentage, %	0.683 **	*p* ≤ 0.001	0.560 **	*p* ≤ 0.001
Δ Fat-free mass, kg	0.621 **	*p* ≤ 0.001	0.480 **	*p* ≤ 0.001
Δ Skeletal muscle mass, kg	−0.614 **	*p* ≤ 0.001	−0.593 **	*p* ≤ 0.001
Δ Waist circumference, m	0.671 **	*p* ≤ 0.001	0.540 **	*p* ≤ 0.001
Δ Waist-to-hip ratio	0.691 **	*p* ≤ 0.001	0.650 **	*p* ≤ 0.001
Δ Adiponectin, ng/mL	−0.526 **	*p* ≤ 0.001	−0.542 **	*p* ≤ 0.001
Δ Leptin, ng/mL	0.603 **	*p* ≤ 0.001	0.454 **	*p* ≤ 0.001

** Significant with *p* ≤ 0.001.

## Data Availability

The data presented in this study are available upon request from the corresponding author.

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
