# Peer review of "Long-Term Resistance–Endurance Combined Training Reduces Pro-Inflammatory Cytokines in Young Adult Females with Obesity"

_sports, 2023, doi:10.3390/sports11030054_

Round 1

Reviewer 1 Report

General comments

The authors have clearly stated that the purpose of the study was to examine the effect of different types of exercise on decreased pro-inflammatory cytokines in obese female adolescents. The paper is well-written, easy to follow and adds merit to the vital role of combined (endurance and resistance) training in health among youth people with obesity. Given this approach, this work can enhance future attempts in similar research area. However, I have highlighted a few suggestions and concerns in my specific comments section (below) that need to be addressed before considering whether this work should be published or not.

 Specific comments

 ABSTRACT

 - Exact percentage of change, p-values, and effect sizes should be presented in results.

INTRODUCTION

 - I suggest adding a sentence about the beneficial role of resistance training in general health of adults according to the latest guidelines by the World Health Organization (1) and the American College of Sports Medicine (2).

- A statement about the popularity of resistance training programs with free weights in the health and fitness industry at European and global level, according to the latest report published by the American College of Sports Medicine (3), it could be a useful addition.

 Suggested references:

1.      Bull FC, Al-Ansari SS, Biddle S, Borodulin K, Bumanat MP, Cardonal G, et al. World Health Organization 2020 guidelines on physical activity and sedentary behaviour. Br J Sports Med 2020; 54(24): 1451-1462.

2.      American College of Sports Medicine; Liguori, G.; Feito, Y.; Fountaine, C.; Roy, B.A. ACSM’s Guidelines for Exercise Testing and Prescription, 11th ed.; Wolters Kluwer Health: Philadelphia, PA, USA, 2021.

3.      Kercher VM, Kercher K, Levy P, Bennion T, Alexander C, Amaral PC, et al. 2023 Fitness Trends from Around the Globe. ACSMs Health Fit J 2023; 27(1): 19-30.

 MATERIALS AND METHODS

 -          Was this study registered as a clinical trial before the commencement of the intervention? If so, you should mention that. Otherwise, explain why you did not register this study in advance at any national or international database in order to raise the credibility and transparency of your work.

-          Add the flowchart of the study (Figure), showing potential dropout.

-          Dropout and attendance rates should be presented at the beginning of the results section.

-          Considering the dropout (if any), please clarify what kind of analysis was used (per protocol or intention-to-treat). In case of a dropout rate ≥20%, an intention-to-treat analysis is recommended given the small sample size. Please revise the results section, including tables and figures accordingly.

-          Figures 1 and 2 are very nice and explanatory.

RESULTS

- Exact percentage of change, p-values, 95% confidence intervals, and effect sizes should be presented in results, including tables.

 DISCUSSION

 - Add the most recent evidence of the effectiveness of various exercise types on several cardiometabolic health-related parameters in obese individuals (4).

- Strengths should be added in brief before limitations in the same paragraph at the end of the discussion section.

- In conclusions, you should underline the main findings and suggest future research attempts in this area while highlighting potential practical implications.

 Suggested reference:

4. Batrakoulis A, Jamurtas AZ, Metsios GS, Perivoliotis K, Liguori G, Feito K, et al. (2022). Comparative efficacy of five exercise types on cardiometabolic health in overweight and obese adults: a systematic review and network meta-analysis of randomized controlled trials. Circulation: Cardiovascular Quality and Outcomes, 15(6), e008243.

Author Response

Dear Reviewer. Thank you very much for your kind consideration, your valuable comments, and your comprehensive suggestions for our manuscripts. The constructive suggestions for this manuscript are crucial to improve the betterment of understanding global academic audiences. According to your positive comments, a brief explanation for your valuable feedback "Please see the attachment".

Yours Sincerely,

Purwo Sri Rejeki and co-authors

Reviewer 2 Report

I would like to thank the authors for this interesting paper. I have the following suggestions that I think could improve the manuscript:

Title: I don’t understand why you use the word proinflammatory (adjective) afther the word cytokines (noun). It should be the other way around (the same throughout the whole text)

Line 26: which will have implications for increased pro-inflammatory cytokines: needs to be rephrased

Line 31: instead of given, probably followed

Line 33-34: no need to describe what method you used to measure the cytokines in the abstract section

Line 48-49: the whole sentence needs to be rephrased

Line 52: either you put first capital letters in all words when presenting and abbreviation or none (the same throughout the whole manuscript)

Line 54: define what is M1 activation

Line 55: such AS

Line 59: Chronic inflammation associated with aging should not be reported here since you are describing inflammation due to obesity

Line 71: Il-6 secretion to be released: needs to be rephrased

The whole last paragraph of Introduction: since the manuscript is not about the molecular mechanisms of how exercise improves metabolism, I would suggest that you keep the relevant literature data discussion to a minimum.

Line 92: what do you mean by “This study was a true experimental study”?

Line 93: Since BMI is 30.93±3.51 kg/m2, how can all women be obese (and not some of them overweight)?

Lines 105-110: Rephrase the sentences - verbs are missing

Lines 166-167: All abbreviations (including CTRL etc) should be put in the same alphabetical order

Line 229: “The analysis of showed”: of what?

Line 251: “stated”: possibly stating

Line 284: “was not regulated in the study”: needs to be rephrased

Lines 299-304: Data mostly irrelevant to the study’s data and scope, should be probably omitted

In general, the Discussion section should be organized in a better way so as to be more relevant to the paper’s data (IL-6 and TNF changes related to type of exercise) rather than, molecular mechanisms involved in the whole process

Line 308: according the Methods section, the participants were not adolescents but rather young adults

Line 310: that WAS carried out

Line 312-313: I would suggest you keep a more humble profile and avoid sentences such as: “the findings of this study still have significant implications for the advancement of science”

Line 320: “all three types of exercise reduce levels of pro-inflammatory cytokines levels including IL-6 and TNF-α”: you should not generalize since you only measured IL-6 and TNF

Line 324: “this study also discovered a significant correlation 323 between pro-inflammatory cytokines and body fat and adipokines.” I would suggest you omit this sentence, since it wasn’t a target of your study and since the relation between body fat and cytokines is a well established fact.

Author Response

(The authors gave the same response as above.)

Reviewer 3 Report

General

The current manuscript looks at the impact of three different forms of exercise on pro-inflammatory cytokines, specifically IL-6 and TNF-α. You found that both pro-inflammatory cytokines were reduced in all three exercise modes, however, the combination of resistance training and endurance training was superior. In addition, you found significant correlations between the pro-inflammatory cytokines and BMI, fat mass, BF %, and several other anthropometric measurements. While I believe you study has merit, I have several concerns that need to be addressed.

COMMENTS

General

Throughout the manuscript, you make several references to your population being adolescent females, however, the average age (21.86+1.39) makes them young adults. This needs to be corrected through the entire manuscript.

Your entire manuscript is based on dealing with obese individuals, however, you average BMI (30.93+3.51 kg/m2) suggest that several of your participants are not actually obese and just considered overweight. You need to either remove the individuals who do not qualify as obese and run your statistics again, or you need to change the entire theme of the manuscript.

Abstract

Your abstract makes no mention of the length of the intervention, which is a significant piece of this study.

Page 1, Line 24: You use sedentary lifestyle and low physical activity in the same sentence. These are redundant.

Introduction

Page 1, Line 44: You use the phrase ‘prevalence rate’. Prevalence is a rate so you don’t need to use the word rate. This occurs a few times in the introduction.

Page 2, Line 62: You need to expand or clarify what you mean by ‘Poor lifestyle’.

Page 2, Line 72: The 1/2 as part of ERK should be subscripted. ERK1/2

Materials and Methods

Page 3, Line 107: You need to explain what the ‘Seca mBCA 554’ is and that it uses BIA to calculate body composition.

Page 3, Lines 112-137: You need to be more specific and add clarity on the exercise protocol. The MIET was 35 mins, how long did it take to complete the MIRT and MICT? Where they all done 3 days a week.

Results

Page 5, Lines 180-186: You report the data in the text, show change in the figures, and also report the raw data in Table 2. Just reference that raw data can be seen in Table 2 and remove all the numbers as it is just hard to read.

Page 7, Lines 203 & 213: The reporting of group differences for cytokines at baseline should be reported earlier and be added to Table 1.

You should also report any changes that occurred pre- and post-training with your anthropometric measurements.

Discussion

Page 8, Line 245-247: You should specify that combined training is better than aerobic and resistance training independently.

Page 8, Line 257: You can delete the first sentence in this paragraph as it’s repeated in lines 260-261.

Page 9: Another limitation is the length of the intervention. Data can’t be extrapolated to any time-frame outside of 4-weeks.

Author Response

(The authors gave the same response as above.)

Round 2

Reviewer 2 Report

Much improved manuscript.

Reviewer 3 Report

Thank you for making all the suggested changes. I know that they were not all simple fixes and you did a nice job in making the manuscript easier to read and have more of a clinical impact. There are still some minor spelling errors that I'm sure will be caught during proof editing.